Spotted hyaena space use in relation to human infrastructure inside a protected area

Belton Lydia E. 1
Cameron Elissa Z. 1 2 3
Dalerum Fredrik dalerumjohan@uniovi.es 1 4 5
1 Mammal Research Institute, Department of Zoology and Entomology, University of Pretoria , Pretoria , South Africa
2 School of Biological Sciences, University of Canterbury , Christchurch , New Zealand
3 School of Zoology, University of Tasmania , Hobart , Tasmania , Australia
4 Research Unit of Biodiversity (UO-SCIS-PA), University of Oviedo , Mieres , Spain
5 Department of Zoology, Stockholm University , Stockholm , Sweden
Vonk Jennifer
Electronic publication date: 2016 Oct 19
Publication date: 2016
Volume: 4
Electronic Location ID: e2596
Received 2016 May 21; Accepted 2016 Sep 14
Copyright: ©2016 Belton et al.
Copyright year: 2016
Copyright holder: Belton et al.
License: This is an open access article distributed under the terms of the Creative Commons Attribution License, which permits unrestricted use, distribution, reproduction and adaptation in any medium and for any purpose provided that it is properly attributed. For attribution, the original author(s), title, publication source (PeerJ) and either DOI or URL of the article must be cited.
License URL: https://creativecommons.org/licenses/by/4.0/

Keywords: Carnivore, Hyaenidae, Crocuta crocutta, Anthropogenic effects, Home range, Habitat selection, Africa, Carnivora, Resource dispersion

Funding: National Research Foundation University of Pretoria Spanish Ministry of Comptitiveness and Economy This study was funded through incentive funding for rated researchers by the National Research Foundation (E Cameron, F Dalerum), a research fellowship from University of Pretoria (F Dalerum) as well as a Ramón y Cajal fellowship by the Spanish Ministry of Comptitiveness and Economy (F Dalerum). The funders had no role in study design, data collection and analysis, decision to publish, or preparation of the manuscript.

==============================
Increasing human population growth has led to elevated levels of human-carnivore conflict. However, some carnivore populations have adapted to urban environments and the resources they supply. Such associations may influence carnivore ecology, behaviour and life-history. Pockets of urbanisation sometimes occur within protected areas, so that anthropogenic influences on carnivore biology are not necessarily confined to unprotected areas. In this study we evaluated associations between human infrastructure and related activity and space use of spotted hyaenas within one of the largest protected areas in South Africa, the Kruger National Park. Home range size was smaller for the dominant female of a clan living in close proximity to humans than that of the dominant female of a clan without direct access to human infrastructure. The home range including human infrastructure was also used less evenly during the night, presumably when the animals were active. Within this home range, a village area was preferred during the night, when the least modified areas within the village were preferred and administration and highly modified areas were avoided. During the day, however, there were no preference or avoidance of the village area, but all habitats except unmodified habitats within the village area were avoided. We suggest that human infrastructure and associated activity influenced hyaena space use, primarily through alterations in the spatial distribution of food. However, these effects may have been indirectly caused by habitat modification that generated favourable hunting habitat rather than a direct effect caused by access to human food such as garbage. Because of the often pivotal effects of apex predators in terrestrial ecosystems, we encourage further work aimed to quantify how human presence influences large carnivores and associated ecosystem processes within protected areas.

Introduction

Human population growth is bringing people into conflict with carnivores at higher frequencies than ever before (Woodroffe, 2000). Carnivores are considered particularly sensitive to human population growth and urbanisation due to persecution, large home range requirements and slow population growth (Cardillo et al., 2004). The rapid expansion of urban landscapes creates both biotic and abiotic changes that negatively impact carnivore populations (Šálek, Drahníková & Tkadlec, 2015). This can cause local extinctions or active avoidance of humans by carnivores (Ordeñana et al., 2010; Schuette et al., 2013).

However, whilst anthropogenic disturbance is classically known for causing population declines (Woodroffe, 2000), some carnivores have adapted to live in close proximity to humans and may directly benefit from the association (e.g., Fedriani, Fuller & Sauvajot, 2001; Contesse et al., 2004; Bozek, Prange & Gehrt, 2007). For carnivores living in close association with humans, several demographic and behavioural changes have been noted, such as alterations in population density (Fedriani, Fuller & Sauvajot, 2001), home range size (Quinn & Whisson, 2005), diet (Newsome, Ballard & Fleming, 2014), and space utilisation (Gilchrist & Otali, 2002). Such changes are often attributed to anthropogenic food supplementing the diet.

An increase in use of anthropogenic food has frequently been associated with a contraction in home range and core area size (Kolowski & Holekamp, 2008; Newsome, Ballard & Fleming, 2014; Šálek, Drahníková & Tkadlec, 2015). However, home range size and use is also dependent on other factors, such as seasonal variation in native food sources. Typically, the season with a lower abundance of native food has coincided with more frequent anthropogenic food use (Lucherini & Crema, 1994; Kolowski & Holekamp, 2008; Pereira, Owen-Smith & Moleón, 2013). In addition, behavioural and physiological differences related to sex (Beckmann & Berger, 2003) and social status (Boydston et al., 2003) may also influence space use and resource exploitation.

Most research on anthropogenic influences on carnivore biology has been conducted in urban environments (reviewed in Bateman & Fleming, 2012). However, areas of elevated human activity also occur inside protected areas (e.g., Gilchrist & Otali, 2002; Quinn & Whisson, 2005), and many anthropogenic factors influencing carnivore space use may also exist inside protected areas. Despite the obvious management implications of the influence of human activities on carnivore space use inside protected areas, studies on anthropogenic influences on carnivores within protected areas are scarce compared to data focusing on urban and suburban landscapes.

The spotted hyaena (Crocuta crocuta) is a large, primarily nocturnal carnivore that lives in social groups (clans) with fission–fusion social dynamics, meaning that clan members are not always spatially co-located at all times. A clan consists of related females and their offspring as well as unrelated adult males (Kruuk, 1972; Mills, 1984; Henschel, 1986). Within a clan the hyaenas are organized by a strict social hierarchy where females are dominant to adult males, and ranks are temporally consistent (Kruuk, 1972).The range of the spotted hyaena covers much of sub-Saharan Africa, from the Kalahari Desert (Mills, 1984) to peri-urban areas in Ethiopia (Abay et al., 2010), although distribution is patchy (IUCN, 2015). The spotted hyaena is known for its opportunistic scavenging (Mills & Hofer, 1998), and the species will readily exploit anthropogenic food (Yirga et al., 2015). In contrast to many species associated with anthropogenic food use, spotted hyaenas are large carnivores that often hunt large prey (Cooper, Holekamp & Smale, 1999). However, their food is often seasonally variable, a trait associated with anthropogenic food use in other species. In southern Africa, the wet season is typically associated with increased prey availability related to the reproduction of prey species (Pereira, Owen-Smith & Moleón, 2013), while the dry season in contrast is associated with drought and elevated animal mortality partially caused by disease (Owen-Smith, 1990; Pereira, Owen-Smith & Moleón, 2013). Hence, environmental factors could influence anthropogenic resource use in this species.

To date, there is very scant information on the effects of anthropogenic resources on spotted hyaena space use, and all accounts we are aware of come from east Africa (Kolowski & Holekamp, 2008; Yirga et al., 2015). For instance, a single source of anthropogenic food associated with a small human settlement had a marked impact on hyaena space use patterns in the Masai Mara National Reserve in Kenya (Kolowski & Holekamp, 2008). Here, we aimed to test if a larger but less concentrated source of anthropogenic resources would have a similar influence on hyaena space use patterns. We monitored the space use of two spotted hyaena females belonging to separate clans inside the Kruger National Park (KNP), South Africa, one inhabiting an area with high levels of permanent human activity and extensive and complex infrastructure. Hence, this clan had direct access to anthropogenic resources, although those resources were spread out spatially rather than located at a single source. The other clan inhabited an area with very limited human activity and almost no permanent infrastructure. The hyaenas in this clan therefore relied almost entirely on native resources.

Because the often clustered distribution of human resources frequently cause carnivore range contraction (Šálek, Drahníková & Tkadlec, 2015), an observation that is related to the general importance of the distribution of critical resources for carnivore space use (Sandell, 1989), we predicted that hyaenas living in an environment with high levels of human resources would have a smaller home range and use it less evenly than hyaenas living without direct access to anthropogenic resources. We also predicted that human influence on space use would be higher in the food limited dry season compared to the wet season. In addition, human infrastructure and associated activity may simultaneously present sites of high resource availability but also of potential disturbance. Extensive human settlements may be beneficial in that they provide more resources which are reliably available, yet they often represent sites that have higher rates of disturbance and potential risks. We therefore predicted that hyaenas would use areas close to human residential infrastructure in relation to the relative trade-off between the resource value and the cost of disturbance. Such trade-offs could lead to temporal avoidance or preference of areas with human infrastructure, where the use of anthropogenic resources is higher during periods of low levels of human activity.

Figure 1 Locations of the three home ranges within the Kruger National Park as well as the annual and seasonal borders for these home ranges.

The Skukuza clan had access to a village area with four unfenced land use types; highly modified, administration, intermediately modified and unmodified areas.

Materials and Methods

Study area

The KNP is situated in the north eastern corner of South Africa and covers almost two million hectares (Fig. 1). This study took place between May 2007 and March 2010 in a 5,000 km2 southern portion of the park. Vegetation in the study area is characterised by woodland with basalt soils dominated by Clerocarya caffra and Acacia nigrescens, with Combretum species on granite soils (Ogutu & Owen-Smith, 2003). Rainfall is seasonal with the majority falling between October and March, with a peak in January and February (Venter, Scholes & Eckhardt, 2003). Average annual rainfall is approximately 650 mm for the Southern section (Venter & Gertenbach, 1986). Mean monthly temperatures range from 7 to 32°C for this area (Venter & Gertenbach, 1986). KNP hosts a diverse array of herbivorous and carnivorous mammals. Prey available for hyaenas in the Southern section of the park include, along with small mammals; impala (Aepyceros melampus), blue wildebeest (Connochaetes taurinus), Burchell’s zebra (Equus burchelli), greater kudu (Tragelaphus strepsiceros), common warthog (Phacochoerus africanus), imbabala bushbuck (Tragelaphus sylvaticus), nyala (Nyala angasii), common reedbuck (Redunca arundinum), waterbuck (Kobus ellipsiprymnus), steenbok (Raphicerus campestris), common duiker (Sylvicapra grimmia) and Cape buffalo (Syncerus caffer). Other megaherbivores such as African elephant (Loxodonta africana), white rhinoceros (Ceratotherium simum), black rhinoceros (Diceros bicornis), and giraffe (Giraffa camelopardalis) are also available, presumably most often as carrion. Impala in particular constitutes a large part of the hyaena diet in KNP (Henschel & Skinner, 1990; Ryan, 2007). Four large carnivores live sympatrically with hyaenas in KNP; African lion (Panthera leo), leopard (Panthera pardus), cheetah (Acinonyx jubatus), and African wild dog (Lycaon pictus).

Data were collected in two areas with contrasting levels of human activity. The Skukuza area included the Skukuza rest camp and staff village area (31°59′E, 25°00′S). Skukuza is the largest rest camp in KNP and hosts up to 300 visitors. It is also the administrative hub for the whole of KNP with a large staff village. In Skukuza, hyaenas had free access to the unfenced staff village consisting of 250 houses, an enclosed staff compound, a golf course, a shop, communal areas, and administrative buildings beside an enclosed area with tourist accommodation (rest camp). The staff area combined with the rest camp covers 4.3 km2 and houses approximately 2,300 staff (Foxcroft, Richardson & Wilson, 2008). Fences around both individual houses and the compound prevented easy access to household rubbish bins. However rubbish bins in communal areas and larger waste collection sites were unfenced. Open gates and damaged fencing also allowed for opportunistic access to other rubbish bins. Hyaenas also had access to the unfenced car park of a picnic site, which contained rubbish bins, and they were able to walk along the perimeter fence of the tourist rest camp. Visitors are required to return to a camp or leave the park by a specific time that varies throughout the year to coincide with sunset and members of staff do not walk in unfenced areas after dark. Animals were therefore able to use unfenced anthropogenic areas after dark with minimal disturbance. In contrast, we also collected data in a neighbouring area (Doispane; 31°25′E, 25°01′S) approximately 20 km away that had limited levels of human activity and the only permanent infrastructure was a building that occasionally was used for short stays by park staff or guests. The Doispane area was at the border of the park and had similar to the Skukuza area access to permanent water. Vegetation in this region of the Kruger National Park is homogeneous (Rutherford et al., 2006). Water access, which is one of the main drivers behind herbivore distribution within the park (Redfern et al., 2003; Smit, Grant & Devereux, 2007), was similar between the two areas and prey densities are relatively homogeneous throughout this southern section of the Kruger National Park (Seydack et al., 2012). Therefore, the main differences between the two areas in terms of resource availability for spotted hyaenas are likely related to the elevated human presence in Skukuza caused by the Skukuza village complex.

Study animals and instrumentation

Each area (e.g., Skukuza and Doispane) was inhabited by one spotted hyaena clan. The clan in Skukuza consisted of 5 adult females, 1 adult male and up to 9 subadult or young adult males and 7 subadult or young adult females. The Doispane clan was substantially smaller and consisted of 3 adult females, 2 adult males and up to 2 subadult or young adult males and 1 subadult or young adult female. Both clans had juveniles present during the duration of the time they were monitored. We monitored the clan in Skukuza from May 2007 to December 2009 and the one at Doispane from May 2007 until August 2008. Monitoring was primarily done at the den locations but also when animals were opportunistically encountered. Observations were partly done for a concurrent study on the influence of human activity on hyaena behaviour and ecology. We monitored the locations of den sites throughout the study by visiting the clans. These visits varied in frequency from daily to once every second week. When a den was not located within sight of a road, we used clusters of relocations from marked animals (see below) at dawn and dusk to indentify likely den locations, which were confirmed by direct visitations. In each clan we had all animals individually recognized based on general characteristics and spot patterns. We scored rank relationships from the outcome of pair-wise interactions.

We fitted one animal in each clan with a collar mounted GPS unit that was tasked to download data through the GSM network (African Wildlife Tracking, Pretoria, South Africa). We selected the dominant female from each clan to create a reliable comparison (Boydston et al., 2003). The social rank was confirmed through behavioural observations of aggressive interactions between clan members. The animals were immobilised from a vehicle by a veterinarian from South Africa National Parks’ Veterinary Wildlife Service department. Both animals were first baited with three pieces of meat, each containing 2 × 15 mg midazolam tablets to enable safer darting. A combination of 4 mg medetomidine hydrochloride and 60 mg Zoletil was then delivered via a CO2-powered dart rifle. An intramuscular injection of atipamezole was administered to reverse the effects of the medetomidine and animals were kept under observation whilst recovering. The female in Skukuza was fitted with her first collar on the 20th October 2007. This stopped working 5th July 2008 and was replaced 24th April 2009. The second collar stopped working on the 19th November 2009 and could not be removed. The collar on the female in Doispane was fitted on the 20th November 2007 and removed July 2011, although we only had access to data from this collar until 6 March 2010. We therefore collected spatial data on the female in Skukuza during the periods October 2007–5 July 2008 and 24 April–19 November 2009 and on the female in Doispane during the period 20 November 2007–6 March 2010. Hence, we collected data on both clans simultaneously for the majority of the time the Skukuza clan had an active collar, and we additionally collected data on an extended time period for the Doispane clan. Although sample size may bias home range size estimates, we have retained our full data record in the analyses to improve the accuracy of the estimated home ranges for Doispane. With the complete set of locations we had sufficient samples sizes in both Skukuza and Doispane to accurately estimate seasonal home range sizes (Supplementary Information 1), so that the uneven sample sizes should not influence any differences between the clans in terms of home range size. Both females were nursing during the time for which each clan were observed, i.e., May 2007 to December 2009 for Skukuza and May 2007 to August 2008 for Doispane.

Research was approved by the University of Pretoria Animal Use and Care Committee (protocol number EC010-07) and the Kruger National Park Animal Use and Care Committee, and was additionally carried out under a research permit from the South African National Parks Board for the project “Impact of human habitation on population dynamics of spotted hyaenas”.

Data collection, classification and analyses

The collars were set to take readings on an 11 h schedule. This schedule provided temporally independent points that covered all hours of the day. Each relocation point was classed as night, day or den. We regarded the time between one hour before sunset and one hour after sunrise as night time and times outside of these hours were as day time as it correspond to spotted hyaena activity patterns (Henschel, 1986; Kolowski et al., 2007). Although we acknowledge that we did not have direct measurement of activity during each of these time periods, our observations confirmed that activity within both clans were principally nocturnal, suggesting that most locations during the night were of active animals and locations during the day were resting locations. Data on sunrise and sunset times for the local area were retrieved from a weather service internet site (http://www.timeanddate.com). However, any relocation that occurred within 30 m of an identified den site was labelled as den points regardless of the time of day. In addition to these three classes of locations, we also grouped relocations by season. Following Venter, Scholes & Eckhardt (2003) we defined all relocations between October and March as having occurred during the wet season and the other relocations as having occurred during the dry season.

We used 95% Minimum Convex Polygons (MCP’s: Mohr, 1947) to estimate home range sizes for each animal. We used MCP’s to quantify home range sizes because they are relatively robust to possible temporal autocorrelation among data and they do not rely on arbitrarily chosen smoothing parameters or spatial resolutions of the underlying reference grid, which could influence the resulting space use estimates (Swihart & Slade, 1985; Row & Blouin-Demers, 2006; Boyle et al., 2009). MCP estimates are also repeatable across different software programs and therefore provide results that are directly comparable with those of other studies (Harris et al., 1990; Larkin & Halkin, 1994; Lawson & Rodgers, 1997). During October 2008, the clan at Doispane shifted its home range to the west with only a small overlap with the previous home-range. This shift included a shift in den locations, and our observations confirmed that all clan members appear to have shifted their movement patterns along with the marked female. We have therefore treated these two areas as separate home ranges for our analyses. Due to their highly clustered nature, we removed den site locations from all home range size estimations, but we have included them in the visual representation of the home ranges because den location potentially can influence home range patterns. For each home range, we created three size estimates, one including all relocations, one for the wet season and one for the dry season. We based our home range estimates on 745 locations (470 in the dry and 275 in the wet season) for the Skukuza female, 269 locations (138 dry and 131 wet season) for the Doispane female in the initial home range and 558 (195 dry and 363 wet season) locations in the subsequent home range.

We used two metrics to evaluate the spatial patterns of utilization within each home range. First, we quantified the utilization of the home ranges during the night, i.e., when we regarded the animals to have been active, using a normalized Shannon spatial diversity index (Payne et al., 2005). This index provides a measure of the evenness of home range utilization and varies from 0, which indicates a completely clustered utilization, to 1, which indicates a completely even utilization of the home range. The index is a quantification of continuous use of space, albeit sampled at discrete points in time. We selected this index for the night time locations because we regard them to be instantaneous point samples of a continuous movement process, and hence this index to be more appropriate than indices that explicitly evaluate patterns of discrete spatial points. To calculate this index, we first created a grid where the cells corresponded to 1% of respective home range, and calculated the number of relocations within each cell. The grid was confined within each respective estimated home range border. We selected this grid resolution as it provided a sufficient number of cells for calculations while avoiding an excessive number of empty cells. We calculated the index H′ as:

H′=∑i=1NPi×lnPilnN

where N is the total number of cells in each home range and Pi is the proportion of relocations in each given cell i. We calculated indices for all relocations combined as well as one index for each season. Second, we used the nearest neighbour index to quantify the spatial distributions of day time locations, i.e., when we regarded animals to have been resting (Clark & Evans, 1954). We opted for a separate index for the day time locations because it is explicitly quantifying the spatial distribution of discrete spatial events, which we believe was appropriate for the distribution of locations when animals were assumed to have been stationary. The nearest neighbour index (R) ranges from 0 (totally clustered distribution) to 2.15 (completely even distribution), and is scaled so that a value of 1 indicates a random distribution, values >1 indicate an over-dispersed distribution and values >1 indicate a clustered distribution. For both indices, we evaluated if the observed values deviated from expectations based on a random spatial distribution of points by generating 1000 random point data sets for each home range, each constrained within the home range border and with the same number of locations as the real datasets, and then calculated the index values for each of these random data sets. A random utilization is a sensible expectation to have under the null hypothesis of no preference for features or areas within a home range (Samuel, Pierce & Garton, 1985). We evaluated how likely the observed index values were under random expectations using a z-score transformation based on values from the randomly generated data (Baddeley, Rubak & Turner, 2015). As a heuristic way of comparing the spatial distribution of night and day time locations between the Skukuza and the Doispane clans, we subtracted the observed value from those calculated from the random data sets (Manly, 2007), and used these deviations from random expectations to compare the Shannon and nearest neighbour index between the Skukuza and each of the Doispane home ranges using two-sample permutation tests. We did one comparison for each pair of home ranges (i.e., Skukuza and each of the two Doispane ranges) for both seasons combined as well as one for each season.

We evaluated the utilization of the urban village area in Skukuza at two separate scales. First, we outlined the whole village area using satellite images retrieved from Google Earth (www.google.com/earth/), supplemented with GPS data collected in the field. We quantified the number of night time (i.e., active) and day time (resting) locations within and outside this area. Second, we described the utilization of different land use types within the village area. For this quantification, we similarly created a map that delineated four different types of land use in the area; highly modified areas—unfenced area with high levels of human use that are unfenced, administration areas—unfenced areas containing business buildings and their surrounding car parks with no fences, intermediately modified areas—areas that have been altered from their natural state but are without buildings or facilities, e.g., golf course and a cricket pitch, and unmodified areas—unaltered habitat inside the village boundary. We then scored each location in the village area to belong to each of these four classes. For both scales, we used a simple resource selection function to determine whether areas were preferred or avoided during night and day, i.e., while active or resting. Following Manly, Mcdonald & Thomas (1993), we calculated the selection indices βi as:

Bi=Wi ∑i=1HWi

where wi is the selection ratios for each land use class i (i.e., the proportion of locations within each class divided by the proportion of available land that each class was covering) and H is the total number of land use classes. For ease of interpretation, we scaled each index so that a value of zero indicates that a class has been used in relation to its availability, a negative value suggests avoidance and a positive value suggests selection (Dalerum, Boutin & Dunford, 2007). We evaluated whether the utilization of the different habitat classes (i.e., within or outside of the village area or the four land use types within the village area) deviated significantly from a utilization based on availability using chi-square tests.

Table 1 Sizes (km2) of seasonal and annual home ranges (95% MCP) in three areas of different levels of human activity.

The low activity sites were inhabited by the same clan that sequentially shifted its home range half way through the study.

Clan	Human activity	Annual	Dry season	Wet season	
Skukuza	High	33.7	31.6	24.5	
Doispane a	Low	53.1	44.4	39.7	
Doispane b	Low	47.9	41.0	45.6	

Results

Home range sizes varied both seasonally as well as between the two females. Despite being part of a larger clan, the female in Skukuza had a smaller home range than the Doispane one both annually as well as within each season (Table 1). The home ranges were not utilized evenly, and all home ranges were less evenly used by night and had more clustered patterns of daytime locations than expected by random distributions (Table 2). The Skukuza female had a different spatial distribution of locations during the night compared to the Doispane female, for both seasons combined (Skukuza vs. Doispane a, Z = 43.8, p < 0.001, Skukuza vs. Doispane b, Z = 44.1, p < 0.001), for the dry season (Skukuza vs. Doispane a, Z = 13.1, p < 0.001, Skukuza vs. Doispane b, Z = 43.2, p < 0.001), and for the wet season (Skukuza vs. Doispane a, Z = 43.4, p < 0.001, Skukuza vs. Doispane b, Z = 41.2, p < 0.001). Although the Skukuza female utilized its home range more evenly than the utilization in the second Doispane range on an annual basis, it utilized its home range less evenly than both Doispane ranges within each season (Table 2). Similarly, the distribution of day time locations differed between the Skukuza and the Doispane females (both seasons combined: Skukuza vs. Doispane a, Z = 6.69, p < 0.001, Skukuza vs. Doispane b, Z = 13.2, p < 0.001; Dry season: Skukuza vs. Doispane a, Z = 32.2, p < 0.001, Skukuza vs. Doispane b, Z = 15.1, p < 0.001; Wet season: Skukuza vs. Doispane b, Z = 33.0, p < 0.001), with the exception of differences between Skukuza and the second Doispane home range during the wet season (Z < 0.01, p = 0.998). The day time points in the Skukuza home range were more clustered than both Doispane ranges for both seasons combined as well as for the dry season, but were more clustered than only one of the two Doispane home ranges during the wet season (Table 2).

Table 2 Spatial distributions of spotted hyaena night and day locations in three home ranges with contrasting levels of human activity.

The low activity home ranges were inhabited by the same clan that sequentially shifted its home range half way through the study. The spatial distribution of active points were evaluated with using a normalized Shannon spatial diversity index (H′), which ranges from 0 (completely clustered use of space) to 1 (completely even use of space). The spatial distributions of resting sites were quantified as a nearest neighbour index (R), which ranges from 0 (totally clustered distribution) to 2.15 (completely even distribution). A value of 1 indicates a random distribution, values >1 indicate an overdispersed distribution and values <1 indicate a clustered distribution.

Home range	Human activity	Night	Day	
		H′Obs	H′Exp	Z	P	RObs	RExp	Z	P	
Both seasons									
Skukuza	High	0.69	0.85	40.3	<0.001	0.41	1.03	16.2	<0.001	
Doispane a	Low	0.71	0.84	14.4	<0.001	0.44	1.04	10.4	<0.001	
Doispane b	Low	0.65	0.77	27.6	<0.001	0.39	1.03	15.2	<0.001	
Dry season									
Skukuza	High	0.63	0.78	30.7	<0.001	0.35	1.03	14.5	<0.001	
Doispane a	Low	0.71	0.91	12.1	<0.001	0.52	1.06	6.88	<0.001	
Doispane b	Low	0.75	0.87	10.6	<0.001	0.42	1.05	8.79	<0.001	
Wet season									
Skukuza	High	0.69	0.85	18.8	<0.001	0.52	1.04	7.96	<0.001	
Doispane a	Low	0.81	0.90	5.90	<0.001	0.55	1.07	5.55	<0.001	
Doispane b	Low	0.70	0.81	16.9	<0.001	0.39	1.04	11.9	<0.001	

For the Skukuza female, more locations during the night were found inside the village area than what could be expected based on its proportion within the home range (Table 3), for both seasons combined (χ2 = 67.4, df = 1, p < 0.001) as well as for both the dry (χ2 = 21.9, df = 1, p < 0.001) and the wet season (χ2 = 50.7, df = 1, p < 0.001). Within the village area, the utilization of the different land use types also differed from their availability (both seasons combined χ2 = 86.3, df = 1, p < 0.001; dry season χ2 = 48.4, df = 1, p < 0.001; wet season χ2 = 47.6, df = 1, p < 0.001), with the intermediately modified and unmodified areas being preferred and the highly modified and administration areas avoided (Table 3). During the day, the village area was neither preferred nor avoided (Table 3; both seasons combined χ2 = 0.59, df = 1, p = 0.443; dry season χ2 = 3.44, df = 1, p = 0.063; wet season χ2 = 0.48, df = 1, p = 0.488). Within the village area, however, unmodified habitat was generally being preferred during the day (both seasons combined χ2 = 17.9, df = 3, p < 0.001; dry season χ2 = 7.27, df = 3, p < 0.063; wet season χ2 = 8.88, df = 3, p = 0.031, Table 3).

Table 3 Spotted hyaena utilization of a village area and of different land use types within this village area in the Kruger National Park.

Percent of locations for the non-village and the village area refer to the percent of all locations within the home range, whereas the percent of locations of each land use type refers to the percent of locations within the village area. Beta coefficients describe relative selection for the village area and within the village area also for the different land use types, scaled so that values <0 indicate avoidance (i.e., that an area is used less than its availability) and values >0 indicate preference (i.e., that an area is used more than its availability).

Land use type	Night	Day	
	Annual	Dry season	Wet season	Annual	Dry season	Wet season	
	% of locations	β	% of locations	β	% of locations	β	% of locations	β	% of locations	β	% of locations	β	
Non-village area	64.1	−0.29	72	−0.21	53.4	−0.27	89.9	0.1	94.1	0.21	82.2	0.05	
Village area	35.9	0.29	28.0	0.21	46.6	0.27	10.1	−0.1	5.9	−0.21	17.8	−0.05	
Administration area	0	−0.25	0	−0.24	33.3	−0.21	0	−0.25	0	−0.25		−0.25	
High impact area	2.27	−0.22	2.20	−0.25	0	−0.25	0	−0.25	0	−0.25		−0.25	
Intermediately modified habitat	50.0	0.30	50.0	0.14	64.4	0.44	4.80	−0.19	0	−0.25	7.70	−0.16	
Unmodified habitat	46.2	0.16	46.0	0.34	32.2	0.03	95.2	0.69	100	0.75	92.3	0.66	

Discussion

Annual and seasonal home range sizes for the Skukuza female were consistently smaller than both of the Doispane female’s home ranges. These observations suggest that human activity and infrastructure were associated with spotted hyaena home range sizes according to our first prediction, i.e., that human infrastructure and activity would be associated with smaller home ranges. Such an interpretation of our results would agree with previous studies that have highlighted that access to anthropogenic areas may reduce carnivore home range sizes (e.g., Šálek, Drahníková & Tkadlec, 2015). Increased availability of resources may reduce home range sizes, especially for larger carnivores that often need to use large areas in search of prey (Kolowski & Holekamp, 2008; Gerht & Riley, 2010; Newsome et al., 2013). Our comparison included two females of equal rank that inhabited areas of similar habitat with comparable prey densities, and therefore we suggest that human infrastructure and activity were associated with spotted hyaena space use by altering resource distributions. We highlight that this interpretation is further supported both by the differences in clan sizes and in number of re-locations for each seasonal home range. The clan with the larger size would be predicted to have a larger home range because of an increased metabolic need (Gittleman & Harvey, 1982), and any potential sample size effect would cause a positive relation between number of re-locations and estimated home range size (Boyle et al., 2009). Instead, we observed the opposite, the clan with the smallest home range was both the largest and had the most number of relocations for home range estimation. We interpret these observations as further support for an association between human activity and infrastructure and the observed home range sizes.

In addition, the Skukuza female utilized its home range less evenly than the Doispane one, which emphasizes that human infrastructure and related activity may not only have been associated with total home range sizes, but also with how hyaenas used space within these areas. Space use was aggregated during both night and day time, which agrees with previous observations of spotted hyaenas (Henschel, 1986; Boydston et al., 2003). For both seasons the Skuzuza female used its home range less evenly than the Doispane one. Patchy resource distributions have often led to uneven space use (Macdonald, 1983; Gilchrist & Otali, 2002). We therefore suggest that the less even space use in Skukuza supported our second prediction, i.e., that spatially concentrated resources associated with the village area would cause a less even home range utilization. Although the day time locations, presumably when animals were resting, similarly were more clustered than random expectations, day time locations in Skukuza were more clustered than only one of the Doispane home ranges, but not the other. This supports an interpretation where resources associated with anthropogenic food influenced spotted hyaena space use, because food distribution should have little influence on the locations where hyaenas spend their resting hours. However, we note that there was only marginal seasonal variation in the differences between clans in terms of home range size and use. This observation contradicts our third prediction, and instead suggests that temporal variation in native food did not alter the association between anthropogenic areas and spotted hyaena space use.

In agreement with other studies (Quinn & Whisson, 2005; e.g., Bozek, Prange & Gehrt, 2007), the Skukuza female showed a preference for the village area during the night, i.e., presumably when active. Within the village area, hyaenas preferred intermediately modified habitat the most, followed by unmodified habitat. Administrative and high impact areas were both avoided during the night. The intermediately modified habitat primarily consisted of open areas such as a golf course, a cricket pitch and various patches of disturbed but un-built land. Contrarily, there was no significant preference for or avoidance of the village area during the day, and within the village area all other habitat classes but the undisturbed habitat were avoided. We suggest that these observations supported also our final prediction, that resources would be utilized according to a trade-off between potential benefits and expected risks. Although we do not have information on direct access to anthropogenic resources, such resources would have been more available in the highly modified areas which were consistently selected against. The unmodified and particularly the intermediately modified habitat instead presented artificial open habitat patches. In particular, the golf course attracted several prey species such as impala and warthog because of its artificially watered vegetation. We believe that this could have generated a habitat patch with aggregations of native prey that additionally was more favourable for hunting than the surrounding woodlands (Mills & Funston, 2003). We therefore suggest that the village area may not have been utilized to gain direct access to anthropogenic resources, but that the preference for the village area was driven by an indirect access to aggregations of native prey that existed in favourable hunting habitat. Such an interpretation is further supported by the lack of habitat preferences for any but the unmodified habitat during resting hours, because areas close to infrastructure that may represent elevated human activity probably were avoided if they were not associated to direct or indirect benefits (e.g., Gerht & Riley, 2010; Riley et al., 2010).

We acknowledge several shortcomings with our study. Our study is preliminary because we have an effective sample size of only one clan. This limits broader conclusions of our results, but none the less provides some insights into further directions for research. Additionally, we compared two clans over somewhat different time periods. This could have biased the results in four principal ways. First, because sample size is related to estimated home range size until an asymptote in number of relocations has been reached, we could have biased our home range size estimates simply because we used different number of relocations for each clan. However, it appears that we had reached an asymptote for all seasonal ranges. Additionally, we had lower sample sizes for the home ranges that were estimated to have been the biggest. Hence, any potential effect of sample size should have strengthened rather than weakened our conclusions. Second, because we collected data in Doispane during periods when we did not collect data in Skukuza, environmental conditions could have caused additional biases in the results. We can not rule out that such biases influenced our data. However, environmental conditions are relatively consistent in the study area, and we did monitor both clans simultaneously during the majority of the time. We therefore regard it unlikely that temporal variation in environmental conditions had a large influence on our results. Third, the two groups differed in both group size and composition. However, as group size is generally expected to cause increased home range sizes in group living carnivores, and we observed a negative relationship between group size and home range size, we interpret also this potential bias to strengthen rather than weaken our data interpretation. Finally, we collected data on only a single female in each clan. Although we attempted to minimize potential biases by marking the dominant female, state dependent differences such as pregnancy and lactation may still have influenced our comparison (Boydston et al., 2003). However, when observed, both females were nursing. We interpret these observations that both females had similar reproductive states throughout the study, although we can not confirm this with direct observations.

To conclude, although this study was based on observations on only two individuals within two clans, it provided valuable insights into the effects of anthropogenic areas on the space use of a large carnivore inside a protected area. Our observations supported that human infrastructure and related activity were associated with hyaena space use, and that these associations at least to some extent may have been related to resource supply, but only indirectly by generating favourable hunting areas. We highlight that further work is needed to explore associations between humans and large carnivores and their related ecosystem processes within protected areas. In particular, we argue that we need to quantify the relative effects of direct provision of food through anthropogenic resources versus indirect provision of food through the creation of favourable hunting habitats, and if such effects alter large carnivore ecosystem function in protected areas.

Supplemental Information

Supplemental Information 1 Estimation of convergence of home range size estimates

To evaluate whether or not we had sufficient sample sizes to estimate seasonal home ranges we created accumulation curves for each clan and season. We created randomized sets of coordinates with increasing sample sizes from 10 relocations up to the actual sample size used for each seasonal range. For each sample size, we randomly drew 100 data sets without replacements from the original sets of coordinates that was utilized to calculate each seasonal home range, and for each random data set we calculated the area covered by a 100% MCP. These areas were plotted against sample size.

Click here for additional data file.

Supplemental Information 2 Raw data on animal locations

Click here for additional data file.

Supplemental Information 3 Shape files of home ranges and human infrastructure

Click here for additional data file.

We are grateful to Daniel Swanepoel and Mariana Venter for assistance with field observations, to the research office at the South African National Parks board for permission to carry out the study, and and to the Veterinary Wildlife Services in Skukuza for assistance with capturing the hyaenas.

Additional Information and Declarations

Competing Interests

Author Contributions

Animal Ethics

Field Study Permissions

Data Availability

The authors declare there are no competing interests.

Lydia E. Belton conceived and designed the experiments, performed the experiments, analyzed the data, contributed reagents/materials/analysis tools, wrote the paper, prepared figures and/or tables, reviewed drafts of the paper.

Elissa Z. Cameron conceived and designed the experiments, contributed reagents/materials/analysis tools, wrote the paper, reviewed drafts of the paper.

Fredrik Dalerum conceived and designed the experiments, analyzed the data, wrote the paper, prepared figures and/or tables, reviewed drafts of the paper.

The following information was supplied relating to ethical approvals (i.e., approving body and any reference numbers):

University of Pretoria Animal Use and Care Committee (EC010-07), the Kruger National Park Animal Use and Care Committee.

The following information was supplied relating to field study approvals (i.e., approving body and any reference numbers):

We received a research permit from the South African National Parks Board to conduct the project ”Impact of human habitation on population dynamics of spotted hyaenas”.

The following information was supplied regarding data availability:

The raw data has been supplied as Supplemental Files.

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
