# Peer review of "Spotted hyaena space use in relation to human infrastructure inside a protected area"

_PeerJ, doi:10.7717/peerj.2596_

## Round 0.1 · original submission · Major Revisions

I have now received three helpful reviews of your MS and have read the MS myself. Two of the reviewers were quite positive. However, the third reviewer raised a number of concerns, which unless rebutted, would preclude your paper from being published in PeerJ.

Like all of the reviewers, I share enthusiasm for the topic of your study and commend you for undertaking the work. I also concur with the reviewers that the MS is clear and readable, so I hope that you can address Reviewer 3’s concerns. It will be especially important to address the validity of generalizing from the activity of a single female to the activity of the entire clan. Reviewer 2 also raised some important questions about access to food sources, collaring and so on that need to be addressed. This reviewer also provided some feedback on an annotated copy of the MS. Please be sure to review those edits and comments. Given that you have only one example of each of your comparison groups (heavy human encroachment, less human encroachment), it is critical that you address issues regarding other differences between the clans other than degree of human encroachment. Were the terrains similar? Was there differential competition for food sources? The collar functioning appears quite different between the two hyaena females. How did this affect the comparability of the data? You indicate that the study took place between May 2007 and Dec 2009 (lines 90-91) but the collar on the Doispane female was on until July 2011 – did you not collect data after 2009? Please clarify. I don’t think that table 1 is necessary. These definitions can be addressed within the text.

Given the potential importance of the findings, I do hope you can adequately address the limitations raised.

·

Basic reporting

I was glad to see a study investigating human-carnviore interactions from the perspective of space use and within a protected area. As the authors note, few studies focus on human impacts within protected areas, particularly impacts on hunting carnivores (e.g. spotted hyenas).
Lines 82-84: The rationale behind the authors' predictions is unclear. For example, what is the basis for the prediction that human activity will decrease home range size? Is it because resources are more readily available, and thus a smaller area is able to support a clan? A few sentences in the intro. explaining each prediction would be helpful.

Experimental design

"no comments"

Validity of the findings

The discussion focused on the Skukuza clan but could also benefit from more description of the Doispane clan findings: how did this clan utilize space, what resources were available to this clan, what characteristics of the clan's habitat could explain their space use?

Additional comments

Overall, a well written article and research that addresses an important issue: human-wild interaction.
Minor corrections:
Line 237: "more evenly that the utilization"--change 'that' to 'than'
Line 244: "The resting points in the Skukuza range was"--should be 'were' not 'was' (the resting points were...)

·

Basic reporting

The manuscript is clear and well organized. In general, it is well written, although I have marked a number of places where there are typos/missing words/tense issues. The previous literature is referenced appropriately, and the article structure conforms to the templates. Tables and Figures are appropriate.

Experimental design

The data described fall squarely in the scope of PeerJ. The research question is clear and well articulated, and relevant to a large number of researchers in ecology and wildlife management. The sample size is small (only two hyena clans) and the number of data points is not as large as it could be...the researchers used an 11 hour cycle for GPS points, which saves collar battery life, but doesn't provide fine-grained detail on location. There are no data on what food resources the animals used, other than speculation from spatial location and habitat use. It would have been great to have camera traps at the garbage bin locations within the inhabited areas, to see if they were utilizing anthropogenic food sources.

The methods are described in sufficient detail to be replicated, and conform with existing ethical standards.

Validity of the findings

The statistical analyses are well described and will be easily replicated by other researchers. The data are available in a repository for future use by others. The conclusions about home range size differences between the clans are clear and unequivocal. I am less convinced by the speculation that the hyenas in the inhabited areas didn't use the garbage bins for food, and would like to see better evidence for that. Perhaps the authors should couch this particular conclusion and say that it is speculative, and deserves further investigation?

Additional comments

The manuscript was easy to read, and I was very interested in the findings! The behavior of large carnivores in human-inhabited areas is of both theoretical and practical importance, and I commend the authors on undertaking this study. Working in the field is never easy, and these data will be used by many of us as a platform upon which we can build further research. Thanks for the opportunity to review your work.

·

Basic reporting

Some sections of the paper include minor faults in grammar.
The introduction is relatively thorough and introduces the topic and context well. However, I do feel that it should be made slightly more clear that my previous work (Kolowski and Holekamp 2008), which is cited in the introduction, does indeed take place in a protected area, and describes space use changes for an entire clan, as opposed to a single individual, exposed to a human food source. In this context, it should be made more clear what additional information or detail is sought in this study. And, if the goal is simply to see if the patterns are replicated in another system, that should be explicitly stated. One difference here is that the clan is exposed to a much larger more diverse human use site, so perhaps that should be framed as the gap in knowledge that is being addressed.

Line 78: Using the term clans here is misleading, since space use of only two individuals was monitored. This should be corrected here and throughout the manuscript, including the abstract. Individual hyenas move within their territory largely independently of other members of their clan. Of course this varies based on a number of factors like social rank, sex and age. But collaring one hyena in a clan is not the same as collaring one lion in a pride for example (where the group nearly always moves together), and it should not be implied that these are the same thing. Some critical basic social organization and movement information is lacking here as well. In particular that these clans are fission-fusion societies, where individuals are hardly ever all seen together, and where animals often move throughout the territory alone or in small groups. No information on clan size is provided. These are critical pieces of information that would allow readers to more critically evaluate the results presented.

Experimental design

There are a number of concerns related to the design of the study that should be addressed, and most have to do with potential biases that have gone unaddressed. These relate in part to the obvious dangers of a very small sample, but also to issues related to rank, reproductive condition, and # of tracking locations.

Lines 134-135: Confirming the dominant individual in a hyena clan is a difficult task and requires months of observation of nearly all individuals in the clan. This may have occurred in this study but the detail is not provided. Please provide some additional information about the extent of knowledge of these two clans including the clan size, and describe in more detail how social rank of the clan was determined conclusively. Similarly, note that my previous work showed that lower ranking animals were far more likely to utilize anthropogenic food. This makes sense as the highest ranking female in a clan can easily take food from any other individual, anywhere in the territory, without conflict. This makes this individual far less likely to have to move far for food, or compromise in food type selection. This issue should be noted in the methods and addressed in the discussion, as I suspect differences in these two clans would be more pronounced if lower ranking animals were investigated.

Lines 141-145: can we assume that the collar of the Doispane hyena worked the entire time it was attached? Also, these two animals were tracked for different periods of time, yet it does not seem that any attempt was made to correct for a potential bias here. For example, conditions in the area in terms of prey or weather could have been dramatically different in 2010 and 2011. Only one of the hyenas was apparently tracked during this period. Were all locations used? Wouldn’t a hyena with a lot more tracking locations have a greater chance of using its territory “evenly”? How many locations were used from each hyena? This should be listed at the beginning of the results section along with the relative proportion of locations from each season.

Lines 152-154: This strategy is problematic. Previous work (e.g. Kolowski et al 2007) shows that hyena activity periods can vary based on their exposure to human disturbance. This is seen in other carnivores as well. Therefore one cannot be confident that all locations in daytime periods were “resting”. Animals not exposed to human activity are very likely to be active in the daytime, particularly in the morning periods. Hyenas often also spend large periods of the nighttime resting. I'm not sure of a reasonable solution, but at least this issue should be addressed in the discussion. A related suggestion is to look at the time of day that the tracked female used various types of human habitats in the village area. This may potentially show that there is consideration given to human activity periods when hyenas select how and when to use these kinds of areas. In Kolowski and Holekamp 2009 it was clear that hyenas only used areas outside of the protected area after dark, even though they became active before this period.

Lines 166-168: The use of MCP to describe home ranges here is hard to justify. First, the locations are 11 hours apart, and this was explained as being selected to reduce temporal autocorrelation, so this should not be a justification of using MCP. In addition, the objectives here are closely related to details about space use in these animals. MCPs are known to often include large areas of unused terrain simply because of how they are drawn and they incorporate no information about intensity of use. At a minimum, the authors should use a 95% Fixed-kernel Home Range estimator. A core area (e.g. 50%) could also be identified and these could be compared between seasons and time of day as well, allowing much higher resolution in the analysis. The paper’s ability to make conclusions is already limited by the sample size of 2 animals. Use of the MCP further limits the resolution of this dataset to unwarranted levels. If the authors are still concerned about autocorrelation, there is a new approach to creating home ranges that incorporates autocorrelation explicitly. See https://cran.r-project.org/web/packages/ctmm/ctmm.pdf.

Lines 171-173: More information is needed here to justify treating these areas as two separate home ranges. Hyena space use can vary widely across years based on den location or prey distributions (for example). In addition, because only one hyena was collared, how can it be known if the entire clan’s territory shifted, of if space use of only this individual shifted? Did the den location also shift to this new location? Where other hyenas being observed in this clan? Did territory marking behavior shift as well? If not, then all the locations should be pooled and considered one home range.

Also, was any attempt made to distinguish between when these two tracked females had cubs at the den and when they did not? Female space use can change dramatically when females have cubs. If this was not taken into account, it leaves open the possibility that different proportions of the tracking time of these two animals were influenced by reproductive behavior. This must be accounted for. Simply excluding den locations (while a good start) does not account for this. Particularly when the evenness of use of the home range is of interest this is critically important. A female with young cubs would exhibit a much more clustered space use pattern than one that does not, or one that has older cubs.

Line 177: You did not know if the animals were active. If you continue using your time of day distinction, something like this should be changed from “…when animals were active” to “nocturnal periods” or “active periods”.

Line 180: It seems that grid cell size here would have a large effect on this index. How was this 1% of home range cell size selected?

Line 188: It seems that the Shannon index was used only for active locations and the nearest neighbor index only for resting locations. If that is correct, please provide some justification or reasoning for this. It’s not immediately clear why the two indices are used instead of one, and what advantage one might offer over the other.

Lines 230-232: this comparison is not necessarily interesting. One would not expect animals to utilize their territory in a random or completely “even” fashion. There is no reason to assume this, and so this test should be removed. The predictions listed in the beginning should be the focus on the analyses, that is for example, that the human environment hyenas would use their home range less evenly, than the wild setting hyenas.

It seems that something like a Logistic Regression Analysis might be far more interesting for comparing use locations with random locations. A suite of variables including habitat type, a range of human use related variables, potential water sources, and indeed den location could be analyzed together in a multivariate framework. The use of these indices seems to oversimplify what could be a far more rich dataset of tracking locations.

Validity of the findings

The reliance on only two individuals to make conclusions about the influence of anthropogenic food sources on space use patterns is problematic, particularly when hyena clan members exhibit such different and individual patterns of movement, and when factors such as reproductive condition, rank and prey distribution have such clear effects on space use. The selection of the same ranked individuals from the two clans is a smart way to control one of these factors, but no mention is made of reproductive condition, or control of timing or number of locations between the two individuals. In addition no information is provided on the distribution or abundance of natural prey in the two areas, or over time. Without these additional controls to address potentially important biases, the results presented cannot be adequately evaluated. It is similarly troubling that the authors refer to the results as describing the entire clan, as opposed to single individuals, and this implies a lack of recognition of the way in which clan members move in hyena society.

Here and throughout, you cannot refer to your results as describing a clan. These results are relevant to two individual hyenas and you cannot assume that the rest of the clan has the same space use pattern.

Line 293: What evidence is there that the selection of these areas was for easier hunting? Were kills made in this area? This seems like speculation and should be listed more clearly as such, unless other evidence is available.

Additional comments

Some additional minor suggestions and points:

Line 77. Add “in this species” to the end of the last sentence here.
Line 119: Not clear what a “skip” is.
Lines 197-202: The comparisons being made here are not clear. Rephrase this sentence to clearly explain what is being compared to what.
Line 207: remove the word “both”

---

## Round 0.2 · Major Revisions

The more critical reviewer from the first round, along with one of the more positive reviewers, was fortunately able to assess the revision of your MS. As this research is not within my primary area of expertise, I rely heavily on the advice of expert reviewers. Although the reviewers continue to commend you for research into this topic, which is undeniably important, and difficult to undertake, there remain a number of significant shortcomings of your approach. Both reviewers have made a number of comments that still need to be addressed.

I am still concerned with the comparability of the two territories and the observations themselves. You don’t provide much information about Doispane. You claim that the main difference between the areas is the elevated human presence. However, this human presence is likely to have an impact on other aspects of the environment as well, including availability of prey, threat of other predators, and presence of water sources, parasites, and many other factors. Conversely, it is likely that the human camp site was chosen because of different facets of various locations. Either way, the territories probably provide different amenities aside from the anthropogenic influence. You indicate that prey densities were similar (Line 173), but how was this determined? Without knowing more about which aspect of the territories might drive differences in behavior it is difficult to draw important conclusions about the impact of human encroachment.

The comparison is also confounded by several other issues, such as clan size and composition and the possibility of nursing/gestation periods for the females being tracked. Given that the clans were observed over vastly different time samples, it is also possible that differing environmental conditions, such as weather, potential drought conditions etc. affected the clans during periods when they could not be directly compared. For example, the female in Skukuza was never observed during Aug – early Oct. and was observed for two ~Oct – July periods whereas the female in Doispane was observed year round for 3+ years. If females were pregnant and nursing during different observation periods, observations may easily be biased by these events. As Reviewer 3 notes, you must address these biases. It is not enough to say that both females were pregnant at some point. Unless they were pregnant during the same time periods, you have a confound. Using only data collected during the same time periods for both groups when directly comparing them is a good idea.
I think that the different use of space within the village area in Skukuza may be one of the more interesting and important pieces of your data. As Reviewer 3 notes, you do not have a controlled before and after human encroachment comparison, so you cannot talk about causal relationships. You do need to indicate clearly what your data adds to the existing database given the shortcomings. We all believe that the data makes an important contribution, but it is your job to place it in context and convince the reader that you can draw meaningful conclusions despite the existence of many confounds. You should emphasize more strongly that the data is preliminary and needs to pave the way for future research. Here, it would be helpful to outline suggestions for future research projects, including the ability to provide control observations for clans of similar sizes in different environments or tracking temporal changes as human encroachment changes.

You cannot conclude that access to an anthropogenic area reduces home range size (line 360-361) because you cannot determine causality given all the other confounding variables present.
Can you explain why areas within the village provided indirect access to easy hunting grounds? (line 402)

I think it is OK to maintain and defend your statistical approach as is.
Minor Issues:

Please do not use “since” unless in a temporal sense. Replace with “because” on line 103, 267, 370, 405.

Please insert a comma before “which” e.g. line 144, 169.

The “or not” on line 299 is not needed.

On line 382, move “only” to before “one of the Doispane home ranges (line 383).

On line 408, move “only” to before “two individuals”

Be consistent in your use of American versus British spelling (e.g. favourable versus favorable)

·

Basic reporting

The rationale behind the authors' predictions is much more clear in the introduction but the manuscript could benefit from the authors' addressing the predictions in the discussion and in the context of their results.
My only other comment is to check comma use and also make the following minor changes:
Line 100: change "less evenly that" to "less evenly than"
Line 187: remove the period after 'and. could not be removed'

Experimental design

Line 167: Can you quantify "several times per week"? For example use the range of days or the average number of days clans were visited per week?

Validity of the findings

Is there evidence that hyenas hunted in the modified habitat more than in the woodland areas or that prey are reliably found in this habitat compared to the woodland? Although this explanation is possible, the study does not address/test the hyena's hunting behavior. Thus, making the argument that hyena's preference of intermediate habitat could be due to "more favorable hunting habitat" needs further justification (e.g. references of studies showing other carnivores preferring to hunt on open golf courses rather than adjacent woodlands?) and explanation or be more carefully couched as speculative and requiring further investigation. Are the resources more available in the highly modified areas? Is it possible that there were sufficient anthropogenic resources to be found in the intermediate habitats? For example, the Skukuza golf course includes facilities that host dinners etc and would presumably be a reliable source of anthropogenic resources. I am not dismissing the authors' explanation of hunting ground access but discussion of alternative explanations would be beneficial.

·

Basic reporting

There are still numerous grammatical errors, mostly having to do with agreement between subjects and verbs.

I specifically suggested broader mention of my previous work in the introduction not to highlight my own research, but to push the authors to state more clearly what gap in knowledge this paper is filling, or what new information it provides. My previous work was associated with a large tourist lodge with multiple buildings, in a protected area, over a large area and looked at the entire clan, over multiple social ranks, in a controlled before/after scenario. While the authors do now list my study, I still feel they should state more clearly how this works compares to that, and what additional questions are being asked. Something along the lines of: “While previous work has shown X in this species, this we aimed to investigate Y.”

As I noted previously, I still feel some mention of the fission-fusion nature of hyena clan social dynamics is warranted. Otherwise readers not familiar with hyenas may assume that these animals are all moving together around their territory.

Table 2 still refers to active vs. resting sites in the caption, as opposed to night vs. day sites.

Experimental design

Lines 102-107: I think the use of the terms “human activity” and “human infrastructure” are here meant to refer to totally different areas or concepts, and that deserves some extra clarification. Whereas the clan exposed to more human activity is predicted to have a smaller range because of human resources, it is stated that hyenas may avoid or prefer areas with human infrastructure. This seems a bit contradictory. Perhaps this sentence should be altered to focus on the timing of use of areas with human infrastructure. Given the trade-off mentioned here, it would be logical to predict that hyenas would minimize potential negative impacts of human interactions by utilizing high human use areas specifically at times when human activity is minimal (i.e. nighttime). Although this night vs. day comparison is a main component of the analysis, it is not mentioned here in the objectives/hypotheses.

It is now clear that one of the clans has 22 animals (Skukuza) and the other only 8 (Doispane). This should necessarily affect territory size of the group yet this is not discussed at all. If anything, this fact makes the conclusions on home range size stronger, because the clan exposed to human activity is larger, yet has a smaller range size. The discussion should include at least some mention of this large disparity in clan size and how it could influence the results, ideally drawing on literature about how territory size may correlate with group size in hyenas. This also implies a much higher hyena density in the human affected area, which is interesting in its own right. I don’t think the authors can actually report on density, not knowing the full territory size of these clans, but it’s worth noting in the discussion. Nonetheless, it should also be noted in the discussion that if the majority of observations on these clans were made at or near the den, then it remains possible that the entire clan was not known, as adult males often spend very little if any time at the communal den, and identification of these individuals requires frequent observation of animals away from the den. It’s not clear from the methods how often hyenas in these clans were observed and monitored away from the den location (aside from the GPS tracking).

Line 172: It’s probably worth at least mentioning somewhere in the paper, perhaps here, that the animals in the clan are organized by a strict social hierarchy where females are dominant to adult males and ranks do not shift over time except for those resulting from births and deaths, and cite a relevant paper. The fact that the authors monitored the two dominant females is very relevant, but most readers may not understand why this is important without a bit more information.

Line 181: What was the goal of the midazolam in the meat bait?

Lines 201-204: I don’t think you need previous citations to define night and day. These citations would only be relevant if you were citing some definition of active vs. resting periods it seems.

The authors have elected to continue using the MCP approach for describing home ranges for the following three reasons: “I) they are robust to autocorrelation, II) they do not rely on arbitrarily chosen parameters (e.g., smoother and size and anchor point of reference grid), and III) they are therefore completely comparable across and within studies.” As their data do not seem to include autocorrelated locations (though this is not specifically analyzed in the paper), the first bullet does not appear to be relevant in this case. For the second point, I would disagree that these parameters are chosen arbitrarily. There are a range of objective approaches for determining the smoothing parameter in a Kernel Home Range (KHR) analysis and while these certainly do have a sometimes large effect on HR sizes, these approaches seem somewhat less arbitrary than the authors’ decision for their grid cell resolution for their own analysis. The authors also state that there are no large areas of the used range that were avoided (and yet included in the MCP) but the reader cannot make this assessment as no tracking locations are pictured in the figures of MCP ranges. I cannot see any reason why the authors would not be interested to do both MCP and KHR based comparisons in this study, given that there are numerous advantages widely accepted in the literature to KHR approaches over MCP (e.g. see Table 5.1, page 134 of Radio Tracking and Animal Populations, Millsapugh and Marzluff 2001). Only additional insight and information can be gained. If the authors do not use KHR, there should at least be some additional discussion of the potential biases involved in comparing MCPs based on different sample sizes. Given that they are heavily influenced by distant locations, MCPs will typically increase as more locations are added until a certain threshold of locations is collected. This threshold is often demonstrated in a graph of # of locations by size of HR, where the HR size should reach an asymptote at some threshold of collected locations. I would prefer to see some demonstration that this asymptote was reached when locations were parsed into wet vs. dry season ranges. In addition, if KHR are not added to the study, I strongly suggest adding some graphical representation of tracking locations, so the readers can see how well the MCP drawn describes space use patterns over the entire tracked period, as well as within seasons.

As noted in the first manuscript review, tracking these single individuals for varying amounts of time and in varying years introduces potential biases. In their rebuttal the authors suggest that this is not an important bias and that any potential differences are “averaged out”. Based on my own experiences, I disagree. One potential option is to investigate the sensitivity of the results to this time bias by only including Doispane locations from the periods when the Skukusa female was tracked to see if any dramatic differences in results are seen. Regardless, any potential biases introduced here by this non-overlapping tracking period must be addressed in the discussion, at least briefly.

The use of the Shannon index approach is still confusing to me, given that Kernel estimators would generate spatially explicit utilization distributions that could be directly overlaid with areas of human activity. The probability of utilization values within each grid cell inside and outside human use areas could be directly compared between night and day locations, and between wet and dry season locations. It seems that this Shannon index approach is an early precursor to kernel utilization distributions and so I’m still unclear as to its advantage in this case, particularly as it is not spatially explicit. The authors recognize this critical difference and note it in their rebuttal. I also still, despite the text added by the authors in this version, do not see any reason to use a different index for day vs. night locations. If a utilization distribution was created, these would be directly comparable between day and night locations. I see that some justification is provided on Lines 266-269, but I think at least some citation should be provided which indicates that either the Shannon Index should not be used to describe fixed locations, or the Nearest-Neighbor should not be used to describe locations from moving individuals. Finally, the authors state that an advantage of the Shannon index is that it allows for statistical comparison to the null hypothesis. I note again below that this comparison is not interesting, but further, a logistic regression framework also allows for a very clear and accepted statistical framework that can be used to assess the importance of habitat features, such as human use areas.

Lines 273-275: Despite the authors’ response, I still do not understand why a comparison with a random distribution (with no underlying ecological data) is an interesting comparison in this context. For an animal so socially focused on its den location, the null hypothesis of even spatial use of the home range is nonsensical. The only interesting comparison, especially as it relates to the authors’ own hypotheses, is to compare these index values between clans. Table 2 and its related results should in my opinion, be removed, unless it is reoriented toward the inter-clan comparisons. Comparison with randomly located points would only be useful in a logistic regression context, when each point was associated with a suite of relevant ecological and potentially anthropogenic variables, such that factors influencing space use could be directly investigated.

Lines 231-237: Spotted hyenas can shift den location regularly. While this will naturally come with a change in space use, this does not necessarily mean that their territory has shifted. Much more common is to have the den location shift within a given territory. While knowledge of a clan’s territory requires multiple years of tracking animals of different social status, the fact that the Doispane clan used these two non-overlapping areas when den use was shifted strongly implies that their full territory size included both the “a” and the “b” home ranges. This would seem to initially suggest that the difference in territory between these two clans is even larger than what was reported. However, the female in the Skukuza clan was not tracked for a long enough period to allow for this type of den location shift to occur, and so it is difficult to make clear conclusions. In any case, the fact that hyenas normally have clearly marked territory boundaries over long periods of time, within which den location can shift, should at least be discussed in this paper, as well as the potential biases this may introduce when comparing only two individuals (one of which was involved in two separate den locations during the study).

Most critically, the authors do not seem to know if these two tracked females were nursing at the time of tracking or not. They state that there were juveniles at the den during the study period, but this is not necessarily relevant. The key question is whether the tracked females were nursing, and how much of the tracking period was contained in these nursing periods. Without somehow controlling this, the results could be highly biased, as there is clear literature showing vastly different space use patterns of females with and without nursing cubs. “Females with den-dwelling cubs had smaller home ranges, were found closer to the communal den, and were found farther from the territorial boundary than were females with no den-dwelling cubs” – Boydston et al. 2003 (Journal of Mammalogy). It’s not clear whether the authors think that cubs are communally nursed (which they are not), but this is the only way the statement that juveniles were in the clan would be relevant. While I agree that, despite my disagreement with the authors about use of MCP vs. KHR approaches, the home range size differences appear clear, one cannot conclude whether this is a result of human activity, as the authors do, without knowing if the reproductive status of these two females was comparable. The influence of this potential bias cannot be overstated. Lines 361-364 could also be explained if the Skukuza female was nursing during more of her tracking period than the Doispane female. Spotted hyena females nurse their cubs for anywhere from about 12 months to fully 2 years. The Doispane animal was tracked for 2 years and 4 months and she could not have been nursing during this whole period. This means she was likely nursing for part of this period, and the same goes for the Skukuza female. But what portion of the tracking periods fell within nursing periods of these two females is absolutely critical. If this is not known, I’m afraid the results are not able to be interpreted with respect to human activity/infrastructure without significant potential unknown bias.

Lines 341-344. I do not understand the authors’ response to my suggestion to use logistic regression. Even if one is only interested in use of human areas, it makes sense from an ecological perspective to account for other factors like habitat and den location in analyzing space use patterns, to see if human use areas have an effect on space use above and beyond these other factors. For example, the results in these lines could strictly be a result of the location of the den during this study period, if it was near the village area. A logistic regression analysis could include a variable like distance to the den, as well as distance to human infrastructure or something similar to see if the den, or the human infrastructure was most important in influencing spatial locations. It’s hard to interpret the results in these lines without knowing where the den location was (please indicate this in the MCP range figures). In this vein I agree with a comment by reviewer #1: “The discussion focused on the Skukuza clan but could also benefit from more description of the Doispane clan findings: how did this clan utilize space, what resources were available to this clan, what characteristics of the clan's habitat could explain their space use?” To date GPS tracking of hyenas has been rare, and this study could be made far more comprehensive and interesting with a more detailed analysis of space use of these two individuals in relation to not only human but also ecological variables, to describe a more complete picture of factors influencing space use.

Validity of the findings

Lines 383-386: Please provide a citation, if there is one, which indicates that food distribution should not influence where hyenas sleep during the day. I would expect that resting locations would be highly influenced by where hyenas are generally spending their time, and data indicates that where hyenas spend their time depends on food, and the den location.

Line 389: And when human activity was relatively low (presumably).

Lines 398-400: This would require that prey species were also utilizing these habitats. Is there any evidence of this? Another interpretation is that hyenas were in fact using the highly modified human areas, searching for food sources, but spent relatively little time there to avoid human interactions. Getting locations every 11 hours would not allow you to necessarily pick up these short bursts of movement into highly modified areas. I do not think you have enough evidence to suggest that “preference” for these human use areas was dictated by hunting opportunities. This is merely speculation and should be clearly stated as such. In fact this was requested already by Reviewer #2 in the first review. Given the low resolution of the movement data, and given previous research on hyena use of human food sources, I believe direct hyena use of human food in these areas (and or the location of the den in this area) are more plausible explanations of the observed space use patterns.

Additional comments

In this version of the manuscript, critical information has now been added which allows further assessment of the validity of the findings including clan sizes, # of tracking locations used for the analyses, and some additional detail on the monitoring protocol during the study period. This is all extremely helpful, yet some potential biases remain unaddressed. And, as in the previous manuscript, there is too little discussion in the paper about the potential dangers/biases in drawing conclusions about broad space use patterns in relation to human activity from this small data set, with unequal effort, in different time periods, in different areas. I generally don’t have any serious issues with the portion of the paper comparing use of grid cells inside and outside the village area, as well comparing use of different subsections within the village area.

I think this is an important data set. There have not been many GPS tracking studies on spotted hyenas in human areas, and this kind of information is critical for long-term conservation/management of this and other carnivore species. The authors rightly have a desire to publish this information and I generally support this. However, it is critical that the relevant biases here are adequately addressed, and due to issues of timing, sample size and a potential lack of knowledge of key parameters of the tracked animals, these biases could be significant enough to make interpretation of the results extremely challenging. The location of the den is extremely important in determining hyena space use patterns and this is not stressed enough in the paper. Removing locations immediately at the den is the right approach, yet den location still is likely to have an overwhelming influence on space use. Clarifying the reproductive condition of the tracking females (nursing or not) is critical.

---

## Round 0.3 · Minor Revisions

Thank you for addressing the comments from the previous round of reviews. The changes have unfortunately introduced some novel grammatical errors (see below). Please correct these in one additional revision so that I can accept the final version of the paper.

On line 91, change “comes” to “come”.
On line 158, change “were” to “was”
Homogenuous should be Homogeneous I think on line 170.
Add “for which” between “time” and “the” on line 214.
On line 221, add “for which” between “the time” and “each clan”.
Delete the “were” on line 239.
Change “since” to “because” on line 265.
On line 413, change “lead” to “led”.
On line 422, change “were” to “was”.
On line 424, change “suggest” to “suggests”.
On line 451 and 455, change “since” to “because”.
On line 452, “move the “only” to before “one clan”.
On line 471, change to “we collected data on only a single female in each clan”.
Add a comma after “observed” on line 474.
It would be wise to change use of the term “influenced” throughout the discussion to something less causally implicative, such as “associated with” or “predicted”. Also please remove the redundant use of “influence in the phrase, “had limited influence on how anthropogenic areas influence spotted hyaena space use” on lines 424-425.

---

## Round 0.4 · accepted · Accept

Thank you for your attention to the last remaining editorial comments. I am pleased to accept your article on such an important topic.

---

## Author Rebuttal · Round 0.4

Dear editor,

We are most grateful for your favourable assessment of our revisions. We have done the requested corrections to this version, in addition to a number of other minor editorial revisions that we found, particularly regarding the formats of the references. All of these are saved using track changes.

Kind regards,

Fredrik Dalerum